

# Mobile phone enabled mental health monitoring to enhance diagnosis for severity assessment of behaviours: a review

Abinaya Gopalakrishnan[1,2], Revathi Venkataraman[1], Raj Gururajan[2], Xujuan Zhou[2] and Rohan Genrich[2]

[1] Department of Networking and Communications, School of Computing, SRM Institute of Science and Technology, Kattankulathur, Chennai, India
[2] School of Business, University of Southern Queensland, Toowoomba, Australia

Corresponding authors
Abinaya Gopalakrishnan,
ag1266@srmist.edu.in
Xujuan Zhou,
xujuan.zhou@usq.edu.au

## ABSTRACT

Mental health issues are a serious consequence of the COVID-19 pandemic, influencing about 700 million people worldwide. These physiological issues need to be consistently observed on the people through non-invasive devices such as smartphones, and fitness bands in order to remove the burden of having the conciseness of continuously being monitored. On the other hand, technological improvements have enhanced the abilities and roles of conventional mobile phones from simple communication to observations and improved accessibility in terms of size and price may reflect growing familiarity with the smartphone among a vast number of consumers. As a result of continuous monitoring, together with various embedded sensors in mobile phones, raw data can be converted into useful information about the actions and behaviors of the consumers. Thus, the aim of this comprehensive work concentrates on the literature work done so far in the prediction of mental health issues via passive monitoring data from smartphones. This study also explores the way users interact with such self-monitoring technologies and what challenges they might face. We searched several electronic databases (PubMed, IEEE Xplore, ACM Digital Libraries, Soups, APA PsycInfo, and Mendeley Data) for published studies that are relevant to focus on the topic and English language proficiency from January 2015 to December 2020. We identified 943 articles, of which 115 articles were eligible for this scoping review based on the predetermined inclusion and exclusion criteria carried out manually. These studies provided various works regarding smartphones for health monitoring such as Physical activity (26.0 percent; 30/115), Mental health analysis (27.8 percent; 32/115), Student specific monitoring (15.6 percent; 18/115) are the three analyses carried out predominantly.

## INTRODUCTION

There are widespread concerns about the impact of the COVID-19 pandemic on mental health, which has led to a natural urgency for mental health research. There are many

contributing factors towards poor mental health, but the COVID-19 pandemic has exaggerated many psychological obstacles, and the hypothesis is that this has subsequently led to a greater incidence of mental health issues in those who may have otherwise never come to the attention of health services (*Zhang & Ma, 2020*). Moreover, the government's order to "stay at home" and the quarantine have led to the longest period of enforced isolation in human history, with the accompanying psychological toll of bereavement and unexpected fatalities. Similarly, the consequences of such widespread isolation have led to a number of intangible effects, such as increased substance use and financial stresses. While the COVID-19 pandemic continues, efforts have turned to early detection of warning markers of psychiatric illness in order to implement time-sensitive therapies (*Adogwa et al., 2016*). Traditional psychiatric evaluation approaches, such as clinical assessments and self-survey reports have limitations in terms of achieving this goal. Firstly, clinical assessments frequently rely on a person's hindsight subjective evaluation of their activities over weeks, months, or years (*Garcia-Ceja, Osmani & Mayora Ibarra, 2015*). As a result, individuals may not be willing to engage appropriately, or may introduce an element of recall bias during the assessment. Secondly, the types of consumers chosen for inclusion in such studies are often sourced from clinical environments and rely on those who are sufficiently disordered to require contact with mental health services. Subsequently, such assessments may have minimal consistency and may be subject to types of reactions associated with a person's drive to undergo therapy *e.g.*, minimising symptoms, hyper-endorsing issues or avoidance of treatment. Furthermore, examinations are frequently conducted after psychological problems or cognitive disability which have progressed to the point where therapy is required for them, as they are much more resistant to treatment. Improved precise diagnosis of behavioural indications linked to imminent difficulties could lead to early remedies, potentially improving long-term outcomes. In the last decade, mobile phones have exceeded their original use as communication devices. A smartphone can now function as a digital camera, accelerometer, activity tracker, or chatbot, amongst other functions. The different embedded sensors, together with the usage of smartphones coupled with widespread availability, have made them a significant research tool in a variety of fields. One such aspect was passively monitoring or self-monitoring for forecasting or categorizing smartphone customers' health-related actions (*Cheffena, 2015*; *Huang et al., 2015*). Manipulation of mobile data such as application usage, communications and performed activities can be converted into latent information for predicting users' well-being. Furthermore, contextual data of users includes weather and Wifi access used to determine location (*Moher, 2009*). The embedded sensors act as effective self-monitoring tools which enable the passive collection through the customized platforms containing microphone, zoom lens, magnetometer, speedometer, bluetooth, light and sound sensors (*Higgins et al., 2011*; *Wang & Zhang, 2015*). This evolution has sparked a lot of curiosity and research possibilities in the context of mental health and wearable technologies. It also emphasises the need for greater research into its mental health implications, based on past material.

The overarching goal of this scoping review is to establish a uniform framework for defining the behaviours collected through various sensing techniques. The specific objectives of this study are:

1. To concentrate on the prospect of employing current mobile or wearable devices to detect and treat mental health issues early.
2. To provide an outline of clinical competencies used to compare the novel findings and data from smartphones that are utilized to track the health of the users.
3. To identify prospective areas for future research in behavioural sciences studies.

This review focuses on evaluating human behaviour during this pandemic using the best companion smartphones and highlights recent advances in the field of mobile phone usability to answer the following primary research questions:

1. Could wearable or mobile tech be used to offer remote psychological assistance during the COVID-19 pandemic recovery process?
2. What are the recent improvements in measuring instruments such as hardware and software?
3. What are the limitations or concerns with passive monitoring that have been detected into the articles integrated in the study are of interest to us?

The rest of the paper is organized as follows. 'Survey Methodology' presents methods for narrowing publications by including the recent works in estimating the mental health issues in this COVID scenario. 'Results and Analysis' deals with the systematic search methods for deducing the nature of mental well-being observing frameworks. 'Discussion' presents different related work for mental health monitoring systems. 'Current Challenges in Passive Sensing for Mental Health Research' describes the significant drawbacks of change in expectation. Finally, conclusions and advanced work are given in 'Conclusions'.

## SURVEY METHODOLOGY

This survey methodology proceeds with a description of the searching strategy, scientific databases retrieved, the inclusion and exclusion criteria, and the number of research articles selected from the various databases to find the research work.

### Search strategy and information Source

The strategic purpose of digitised assessments for psychological health issues has been immediately required during the COVID-19 pandemic. Wearable gadgets can be used to enhance assessments of mental health issues and can be used for tracking at-risk and quarantined populations. Furthermore, while passive monitoring does not operate as frequently as active monitoring, it can collect and generate vast amounts of data, which may be used in supporting clinical assessment. As a result, the search focused on identifying the most appropriate keywords for collecting recent papers on the subject such as mobile technologies, ambient sensing, sensors and wearable and clinical competencies. Searches were conducted in electronic bibliographic databases such as PubMed (Health Science), IEEE Xplore, ACM Digital Libraries (computing methods), Soups (information article), APA PsycInfo, and Mendeley Data (Physio informatics), addressing this _via_ query-based searches.

1. Scopus (ScienceDirect)((ALL (Monitoring deceives)) OR (ALL (Passive monitoring)) OR (ALL (hardware)) OR (ALL (Software)) OR (ALL (smartphone deceives)) OR (ALL ('monitoring AND deceives' OR 'Passive AND monitoring' OR 'hardware AND software' OR 'smartphone AND deceives'))) AND (ALL (well-being * OR physical * OR mental * AND health))

2. IEEE Xplore digital library(''Full Text Only'': mental health monitoring) AND (''Full Text Only'': hardware) OR (''Full Text Only'': software) AND (''Full Text Only'': passive monitoring) AND (''All Metadata'': mental behaviours) AND (''All Metadata'': well being)

3. APA PsycInfo Any Field: mental health monitoring using hardware OR Any Field: software with passive sensing

4. Mendeley ALL(mental AND health AND well AND being AND using AND passive AND monitoring AND sortBy = publicationYear)

5. PubMed((''mental health''[MeSH Terms] OR (''mental''[All Fields] AND ''health''[All Fields]) OR ''mental health''[All Fields]) AND (''health''[MeSH Terms] OR ''health''[All Fields] OR ''well''[All Fields] OR ''well being''[All Fields]) AND (''passive''[All Fields] OR ''passively''[All Fields] OR ''passives''[All Fields]) AND (''monitor s''[All Fields] OR ''monitorable''[All Fields] OR ''monitored''[All Fields] OR ''monitoring''[All Fields] OR ''monitoring s''[All Fields] OR ''monitoring, physiologic''[MeSH Terms] OR (''monitoring''[All Fields] AND ''physiologic''[All Fields]) OR ''physiologic monitoring''[All Fields] OR ''monitor''[All Fields] OR ''monitorings''[All Fields] OR ''monitorization''[All Fields] OR ''monitorize''[All Fields] OR ''monitorized''[All Fields] OR ''monitors''[All Fields])) AND (2003:2022[pdat])

6. ACM [All: mental health behaviors prediction using passive monitoring] AND [All: mental health well being using passive monitoring]

Various reputable databases are excluded so the risk of duplication of research is minimized, for example, the Web of Science database.

### Criteria for inclusion and exclusion

This study mainly focuses on the passive monitoring system for analysing the behaviours to predict the mental health which does not need user interaction, moreover, it is not necessary for monitoring users to communicate with the sensors voluntarily for sensing purposes, it should be embedded as they use in their routines. Thus, articles that were published between January 2015 and December 2020, written in English and meet the above purpose, are included.

In contrast, records excluded are based on the following reasons removal of duplication-429 (sample article indexed in various databases), based on: the title of studies; 192 (active monitoring methods, self-questionnaire methods); irrelevant review objectives, 79; and finally based on the articles that lack availability of results and discussion, 128 (EEG, ECG monitoring). Books, book chapters, conference abstracts, short surveys, editorials and letters were excluded as well. Finally, omitted studies which do not contain proper ethical clearance to carry out the research work.

### Study selection

Studies were extracted based on the query formed above and refined based on their titles, followed by their abstracts. The rest of the refined articles were further scrutinised by reviewing the content and determining relevance to the research question. The search list was ordered chronologically and then analysed by a group of researchers in unison so as to minimise selection bias.

Figure 1 shows the data flow diagram explaining how the publications for the review were chosen based on elimination carried out manually with inclusion and exclusion criteria. The systematic search described above yielded a sample of 943 manuscripts.After removing the duplicates, we were left with 514 studies. 349 of the 514 titles examined were deemed eligible for abstract evaluation. An estimated 199 abstracts were considered as potential review candidates, which led to 182 full-text retrievals and evaluations. Finally, 115 manuscripts were included in the study. Table 1 shows the number of returned and selected papers from retrieved databases.

Based on the exclusion criteria Table 2 shows the count of papers that were excluded from the final stage of the review. Table 3 shows the number of articles published per year in the last six years.

## RESULTS AND ANALYSIS

### Summary of the search results

Tables 4 and 5 provide the summary of various behavioural health outcomes from sensors, wearable remote monitoring intervention studies which include duration of the evaluation, target population, methods involved, resultant outcomes, clinical outcomes, hardware, software components used for monitoring.

From Figs. 2 and 3 and Tables 4 and 5 it can be seen the majority of review articles considered for analysis chose the accelerometer and GPS as a source of the user's data can be seen. The accelerometer provides added benefits such as covered distance, speed, static/inactive and time periods of movements which also provided amiability, movement and confinement. Moreover, the GPS provides dynamic, location variance, entropy, circadian movement and universally common latitude and longitude data from the satellite to predict the intensity of movement. It can also predict sociability, and detachment that can provide context information to predict the mental health issues more accurately.

### Analysis of the results

The framework for the health assessment system represents in Fig. 4 was created to guide the analysis. This framework includes the combination of embedded sensors values used to deduce the majority of behavioural classifiers (markers) to forecast the inferences such as mental health issues, academic performance, fall detection for elderly people, isolation from the social group and even for lifestyle recommendations. Those behavioural markers such as physical activities, usage of the phone, sleep, location and social activities are detected from the sensors data. Finally, inferences are compared with the self-reports as described in Table 6 (ASRM, SGABS, GABS, IBS, HADS, PHQ, GHQ) which are generally used by clinical experts to analyse the health issues. This section deals with a detailed

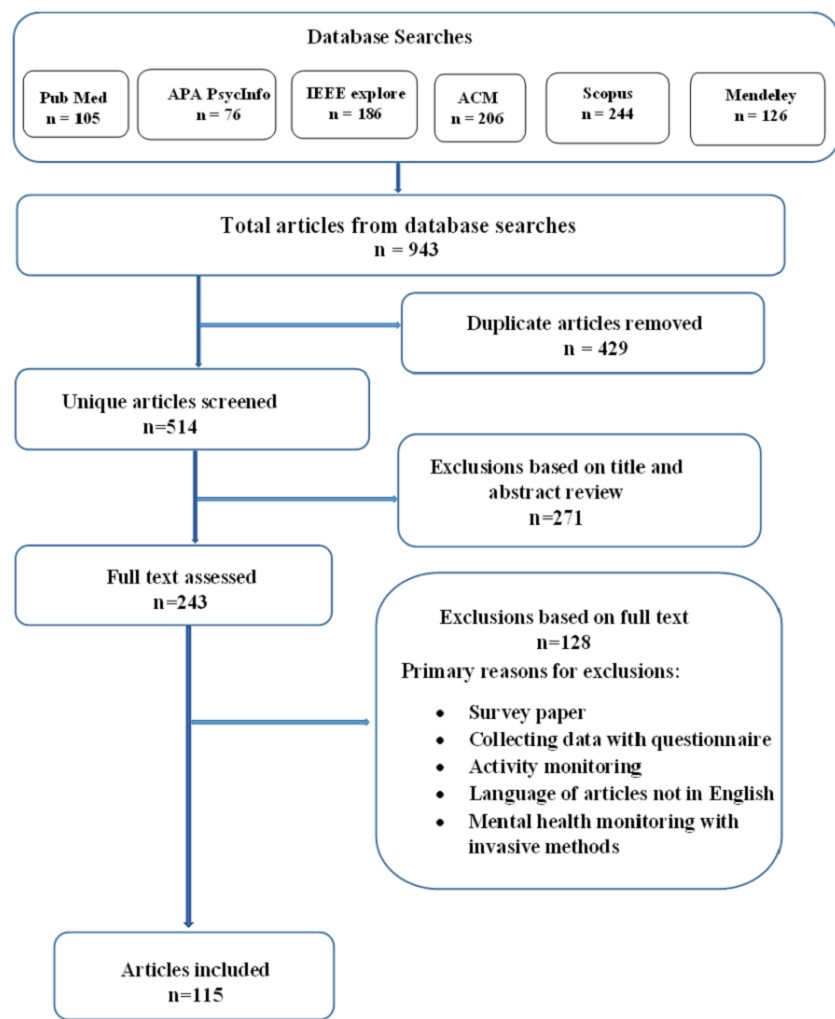

**Figure 1** **Data flow diagram explaining how the publications for the review were chosen.**

**Table 1** **The number of documents resulted and chosen for review from various databases.**

| Articles | PubMed | APA PsycInfo | IEEE explore | ACM | Scopus | Mendeley |
|---|---|---|---|---|---|---|
| Result of the search query | 105 | 76 | 186 | 206 | 244 | 126 |
| Taken for the survey | 30 | 9 | 28 | 20 | 16 | 12 |

review of the embedded sensors and their function, which can result in monitoring various health metrics like behaviour monitoring, physical activities, overall well being, sleep quality, student monitoring and conviviality. In addition, traditional methods used by practitioners were later combined into scales given as self-reports for mental health monitoring.

**Table 2  The distribution of rejected papers as a result of the full-text review.**

| Reason for exclusion | Publications excluded |
|---|---|
| Survey paper | 46 |
| collecting data with questionnaire | 39 |
| Activity monitoring | 12 |
| Language of articles not in English | 2 |
| Mental health monitoring with invasive methods | 29 |

**Table 3  Year-by-year breakdown of the number of unique papers returned.**

| Year | Articles per year |
|---|---|
| 2020 | 19 |
| 2019 | 16 |
| 2018 | 20 |
| 2017 | 15 |
| 2016 | 25 |
| 2015 | 20 |

## Embedded sensors in smartphones

A smartwatch resembles a wristwatch in appearance, but it can do much more than keep time. Bluetooth is supported on digital timepieces, and the functions can be extended to smartphones. In such instances, the smartwatch can be used to read messages, accept calls, monitor the climate, and many more advanced functions.

In addition to these advantages, smartwatches aid in the analysis of the wearer's behaviour and the determination of their mental health. The researchers used a combination of sensors and wearable technology (Table 7). Smartphones (71/115, 61.7 percent), wristbands or smartwatches (44/115, 38.2 percent), were the most commonly utilised gadgets in the research of behaviour monitoring. Accelerometers (46/115, 40.0 percent), mobile phone usage (36/115, 31.3 percent), Global Positioning System (33/115, 28.6 percent), actigraph (23/115, 20 percent), microphones (28/115, 24.34 percent), electrocardiogram (ECG) , and Electroencephalogram EEG (were the most commonly utilized sensors (23/115, 20.0 percent).

From Table 8 it is clear that 72.1 percent of research papers (83/115) used this sensor to collect data from users, most of which were related to physical activity. In smartwatches, the accelerometer is used to detect the wearer's movement and orientation. Regardless of whether one or two hands are used, the accelerometer in the smartwatch detects about two dozen movements and activities (*Lindner et al., 1999*). The controls for software applications are then mapped to these motions. The Tri-axial version is a common smartwatch modification that maintains track of the wearer's physical activity. The Tri-axials record up and down, side-to-side, and back-and-forth movements, unlike the uniaxial version, which only records up and down movements (*Carbonell, Michalski & Mitchell, 1983*).

**Table 4  Summary of behavioral health outcomes from sensors, wearable, and remote monitoring intervention studies.** P- Participants/Study length, CA- Clinical assessment, H/w-Hardware, S/w-Software.

| Study | Period | Population | Method/Outcomes | CA | H/w, S/w | Sensor |
|---|---|---|---|---|---|---|
| *Ponzo et al. (2020)* | 262/4 weeks | College students | BioBase application was used for 4 weeks to reduce anxiety and promote well-being. | STAI, PHQ, WEMWBS | SP (iOS), Wristband (BioBeam), Biobase app | Accelerometer, actigraph |
| *Doryab et al. (2019)* | 160/4 weeks | college students | To detect the loniesss, keep an eye on social and sleeping habits. With an accuracy of 80.2%, it can detect loneliness and changes in loneliness levels, and with an accuracy of 88.4%, it can detect changes in loneliness levels. | UCLA | SP, Wristband, AWARE app (freeware data collection app) | Accelerometer, actigraph, Bluetooth, phone usage, GPS, microphone, SMS usage |
| *Sano et al. (2018)* | 201//4 weeks | college students | Critical items detected using wearable sensors like temperature, barometer such as routine behavior, socializing for stress, depression with 78.3% accuracy for segregating stress level among students. | ASRM, IBS | SP, wristband (Afectiva), Motion Logger (AMI), Funf open-sensing framework | Accelorometer, actigraph, temperature sensor, GPS, light sensor, phone usage |
| *Demasi, Aguilera & Recht (2016)* | 44/8 weeks | Healthy adults | Change over and abnormality in sleep, length of sleep are used to predict emotional wellbeing. | BDI, PHQ-9 | SP (Android), Funf opensending framework | Accelerometer, actigraph, Bluetooth |
| *Gaggioli et al. (2014)* | 121/5 weeks | Healthy adults | Participants reported a signifcant increase in the emotional support skill | COPE-NIV, PHQ, SWLS | SP (iPhone), Wireless cardiovascular belt, body worn wireless sensor | Accelerometer, Bluetooth, Camera, ECG, electrodermal sensor |
| *Knight & Bidargaddi (2018)* | 120/8 months | open | When comparing self-reported data from activity tracker applications to wearables for psychological anguish/moderate level of psychological distress, wearable devices had considerably longer daily activity duration than smartphone apps. | DASS-21 | SP | Accelerometer, actigraph |
| *Szydlo & Konieczny (2016)* | 25/2 weeks | Outpatient | The smartwatch recognises 75% of archetypal ASD motions after six sessions of use with an electronic photographic activity programme. | None identifed | SP (Android), Smart- watch | Accelerometer, actigraph |
| *Garcia-Ceja et al. (2018)* | 30/6 weeks | Healthy adults | Stress detection and prediction using accelerometer data with 95% accuracy | None identifed | SP, Wireless Sensor Data Mining (WISDM), chest sensor, wrist sensor | Accelerometer, actigraph, Bluetooth, microphone, Wi-Fi |
**Table 4** (*continued*)

| Study | Period | Population | Method/Outcomes | CA | H/w, S/w | Sensor |
|---|---|---|---|---|---|---|
| *Huang et al. (2016)* | 16/10 days | Students | Examine the relationship between university students' visits to religious sites and their social anxiety. | SIAS | SP | Accelerometer, GPS |
| *Wang et al. (2016)* | 21/9-36 Weeks | Outpatient | Use random forest regression to correlate smartphone data with schizophrenia symptoms/Significant association between ground truth and anticipated mental health status scores | EMA (measuring sleep, calm, depression, hope, cognition, thoughts of harm, psychotic symptoms) | SP (Android), CrossCheck app, Funf open sensing framework, MobileEMA System | Accelerometer, app usage, GPS, light sensor, microphone, phone usage, SMS usage |

In order to enhance the prediction of behavioural characteristics, context-based scenarios have to be added by using location data from the GPS in devices. The user's location and movements are tracked with this GPS data in most studies 37.39 percentage (43/115). It can be either used alone as an individual sensor or combined along with Bluetooth and Wi-Fi networks. The level of sociability among the users can be easily measured in addition apart from the location measured from the Bluetooth. To determine the wearer's orientation and angular velocity, gyroscope sensors are used. Gyroscope sensors are more advanced than accelerometers in terms of functionality. These sensors are capable of tracking both lateral and tilt orientations. Accelerometers, on the other hand, can only track linear motions (*MacInnes, 2003*). A revolving disc called the motor is mounted on a spinning axis in the gyroscope's design. With the help of Earth's gravitational field, this sensor detects the wearer's orientation (*Liu et al., 2018*).

Out of the refined papers, SMS and calls were used to analyse the depressed mood and social avoidance in (13/115) 11.3 percent of papers, gyroscope and microphone were used for passive sensing in seven percent (8/115) and 5.2 percent (6/115), respectively. The major work of a microphone is to communicate with the opposite parties. By analyzing this factor, the predictions that can be made are mood (modulation of your voice), drowsiness and isolation, while a gyroscope is used to measure the basic day to day activities.

Apart from the data received through sensors, some other additional information can also be collected by the utilization of user patterns for handling the mobile. The communication made using the phone such as calls, text messages, as well as usage of devices incidents such as screen ON/OFF, time spent on the phone, lighting and settings of the device, are used to detect health-related data. Application usage like ambient light and phone screen on/off status were considered as a key factor in 7.82 percentage (9/115) selected publications to predict the sleep of the users. Table 8 illustrates the rest fitness data, such as the camera, and magnetometer, which can be collected from embedded sensors and mobile phones.

The ability to collect data passively is one of the key benefits of using smartphones for health monitoring. All sensor data comes from the smartphone's omnipresent sensors, and passive data collection means there is no user interaction or participation. To detect physical activities, 52 of the 115 publications employed data from only two or more

**Table 5** Summary of behavioral health outcomes from sensors, wearables, and remote monitoring intervention studies. P- Participants/Study length, CA- Clinical assessment, H/w-Hardware, S/w-Software.

| Study | Period | Population | Method/Outcomes | CA | H/w, S/w | Sensor |
|---|---|---|---|---|---|---|
| *Hartanto et al. (2015)* | 5/8 weeks | Healthy adults | Feedback based motivational methods used to predict anxiety with virtual captions | IPQ, SUD | Head mounted VR, Zephyr HxM HR device, Memphis VR dialogue system | Accelerometer, ECG, microphone, Wi-fi |
| *Ben-Zeev et al. (2015)* | 47/10 weeks | Healthy adults | Predicts the correlation between location, length of sleep and stress levels from smartphone data. | PHQ-9, PSS, UCLA-LS | SP (Android), Wristband (JawBone Up), cell towers, wi-fi receiver | Accelerometer, actigraph, Bluetooth, GPS, light sensor, microphone |
| *Lim et al. (2012)* | 537/3 months | Outpatient | Evaluate association of depression using Used generalized estimating equations (GEE) | SGDS-K | SP, Sensor FH62C14 | GPS |
| *Osmani (2015)* | 12/12 weeks | Inpatient, depression and bipolar disorder | The correlation between daily intervals' activity scores and mental state assessment scores was 0.6248, indicating that the mood state (manic, depressed) could be recognised. | BSDS, HDRS | Wristband activity tracker | Accelerometer, GPS, microphone, phone usage |
| *Saeb (2015)* | 28/2 weeks | Outpatient | Predict depressive symptoms/Signifcant negative correlations between GPS features and depression; | PHQ-9 | SP (Android), Purple robot app | GPS, phone usage |
| *Canzian &Musolesi (2015)* | 28/10 weeks | Outpatient | The mobility trace characteristics were linked to depressive mood in a model designed to predict changes in depression based on mobility patterns. | PHQ-8, HADS, GHQ | SP (Android), MoodTraces app | GPS |
| *Grünerbl et al. (2015)* | 10/12 weeks | Outpatient | With 97% precision and 97% recall, detect state shift in persons with bipolar disorder; recognise state with 76% accuracy. | ADL, HAMD, | SP (Android), tracking app | Accelerometer, GPS, microphone, phone usage |
| *Wahle et al. (2016)* | 37/2 weeks | Outpatient | A SVM predicts depression with 61% accuracy, and an RF classifier predicts depression with 59% accuracy. | PHQ-9 | SP, Mobile Sensing and Support (MOSS) app | Accelerometer, GPS, phone usage, SMS usage |

**Table 5** (*continued*)

| Study | Period | Population | Method/Outcomes | CA | H/w, S/w | Sensor |
|---|---|---|---|---|---|---|
| *Maxhuni et al. (2016)* | 10/12 weeks | Outpatient | Ability to classify mood with confdence (85%) in the course of mood episodes | HAMD, YMRS | SP (Android), MONARCA app | Accelerometer, Bluetooth, GPS, microphone, phone usage, Wi-fi |
| *Faurholt-Jepsen et al. (2020)* | 129/9 months | Outpatient | To compare differences in depressed and manic symptoms, researchers used an SP-based method with traditional treatment. | ASRSM, BDI | SP (Android), MONARCA II | Accelerometer, actigraph, GPS, phone usage, SMS usage |

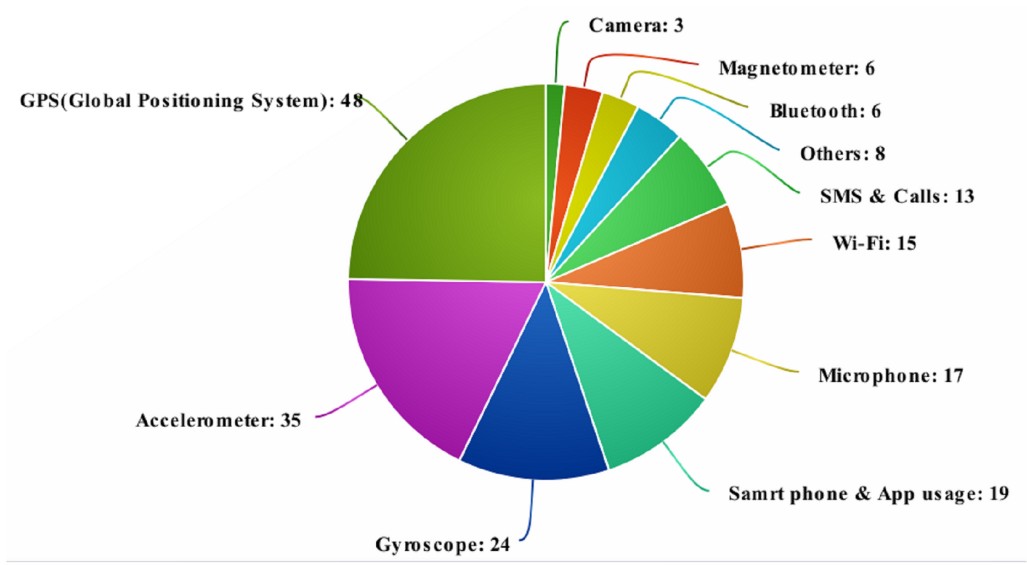

**Figure 2** **Source of health-related data from various sensors.**

**Table 6** **Clinical assessment scales.**

| Acronyms | Description |
|---|---|
| ASRM | Atlman Self-Rating for Mania |
| SGABS | Shortened General Attitude and Belief Scale |
| GABS | General Attitude and Belief Scale |
| IBS | Irrational Belief Scale |
| HADS | Hospital Anxiety and Depression Scale |
| 28 –GHQ | General Health Questionnaire-28 |
| BDI | Beck Depression Inventory |

sensors, the majority of which were accelerometers. Various articles additionally used GPS, camera, and speedometer on their own as described in 115 articles involving the usage of several sensors. In the majority, 23 articles predominantly use the accelerometer, and in conjunction with GPS, Wi-Fi, gyroscope sensors and microphone to determine the physical activities and overall general behaviour.

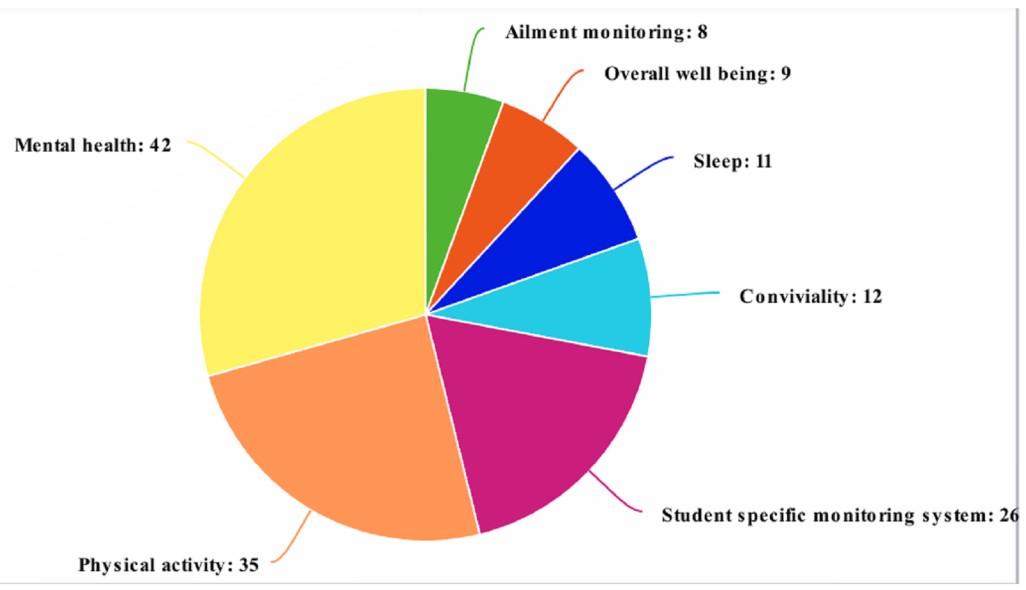

**Figure 3** The selected publications in research fields.

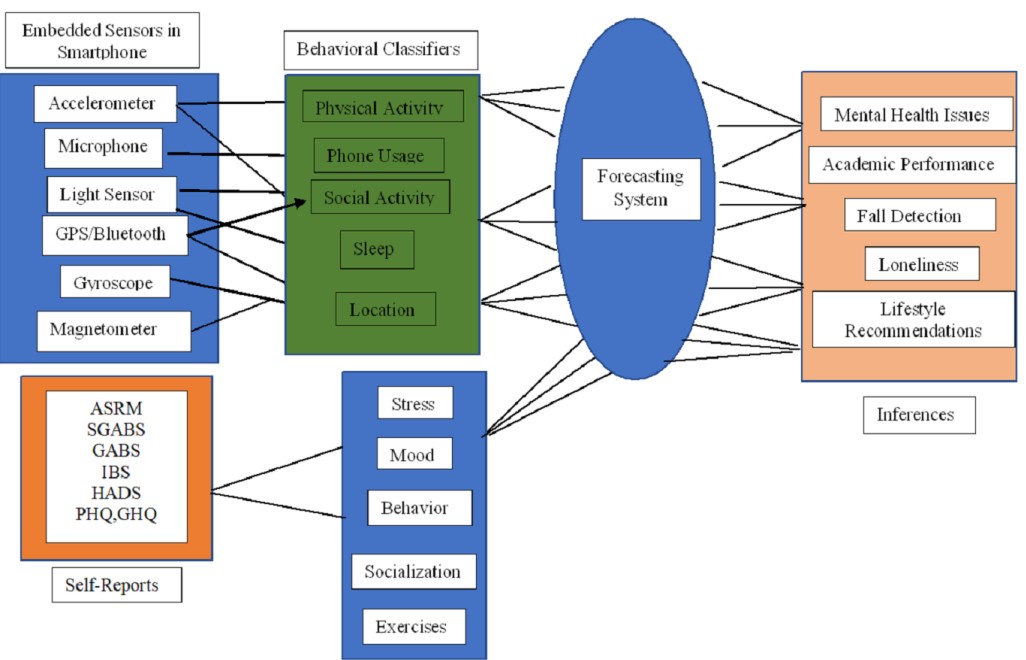

**Figure 4** Framework of health assessing system.

Another important issue that affects the accuracy of the collection of sensing data in the monitoring systems is the operating system. Building for Android and IOS, the two most popular phone operating systems, has its own set of issues and problems. Android is presently the most widely used operating system, with the benefit of being simple to

**Table 7  Passive monitoring sensors.**

| Sensors | Analytics |
|---|---|
| Accelerometer | Covered distance, speed, static/inactive and time periods of movements |
| Actigraph | Physical progress, activity - rest cycles, circadian-rhythm cycles |
| Barometer | To measure the air density |
| Bluetooth | Identify adjacent Bluetooth enabled devices |
| Blood pressure monitor | Systolic and diastolic cycle of the blood, Eye gaze, light |
| Electro Cardio Gram (ECG) | Heart rate activity, heart rate variability |
| Electro Encephalon Gram (EEG) | Monitoring the brain activity |
| Global positioning system (GPS) | Location, duration of the movement, speed, proximity |
| Gyroscope | Gyroscope rotation of the device |
| Light sensor | Measures surrounding and device light |
| Magnetometer | Direction, field strength |
| Microphone | Speech communication |
| pH monitor | Stomach acid secretion intervals |
| Temperature sensor | Skin and ambient temperature |
| Wi-Fi | Location and signal strength of networks |

**Table 8  Source of the health-related data.**

| Components of mobile device used for sensing | No of studies | Low level features | High level behavioral makers |
|---|---|---|---|
| GPS (Global Positioning System), Bluetooth, Wi-Fi | 43 | Movement intensity, Location | Hedonic activity, Stress, Social avoidance |
| Accelerometer | 32 | Activity type, Movement intensity | Psychomotor activity, Fatigue, concentration/Distractibility |
| Gyroscope | 8 | Movement intensity | Hedonic activity, Stress |
| Microphone | 6 | Paralinguistic information, Acoustic environment, Bedtime/Wakeup time | Depressed mood, Stress |
| Camera | 4 | Pictures | Social avoidance |
| SMS & calls | 13 | In phone social activity | Social avoidance, Depressed mood |
| Others (Ambient light, phone screen (On/Off) | 9 | Acoustic environment, Bedtime/Wakeup time | Depressed mood, Stress |

write (*Wang & Zhang, 2015*). The quicker sensor scanning rates of this operating system have been demonstrated (*Boonstra et al., 2017*). Furthermore, IOS prohibits third-party apps from operating in the background indefinitely, preventing data collection (*Hossain & Poellabauer, 2016*). 73.6 percent (67/91) of the chosen papers developed their solution exclusively for Android devices, 18 for both Android and IOS, and six did not specify which operating system they used.

Many of the publications identified using the phones' sensors because of efficiency in energy and reduced overhead in collecting data from users in the passive matter.

**Table 9  The selected publications mental health monitoring.**

| Exploration area | Counts |
|---|---|
| Mental health | 32 |
| Physical activity | 30 |
| Student-specific monitoring systems | 18 |
| Conviviality | 11 |
| Sleep | 9 |
| Overall well-being | 8 |
| Ailment monitoring | 7 |

Advancement in various platforms in machine learning fields provides various tools to develop prediction models to analyse and interpret data in studies using more than three sensors.

## Statistical analysis

Physical activities and mental health, as well as sociability, academic achievement monitoring, and overall well-being, are the most examined health features among the publications chosen for this systematic review, as shown in Table 9.

### Behaviour monitoring

We can group the studies based on the behaviour monitoring from the selected publications and their indicated survey topics. In a few papers, gadgets are used to predict the users' personal life to enhance their monitoring and maintaining their lifestyle such as motor skills of hands in Parkinson's disease, predicting the abnormalities that arise due to sleep in Schizophrenia, abnormal brain neuron activities in Autism and anxiety disorders. People suffering from mental health difficulties such as depression or schizophrenia accounted for 5.2 percent (6/115) of studies with a specific population (*Pratap et al., 2019*; *Zhang et al., 2018*; *Wahle et al., 2016*; *Soares Teles et al., 2017*; *Ng et al., 2014*; *Staples et al., 2017*).

### Physical activity

The papers that were chosen based on the detection of users' physical activity accounts are 26.0 percent (30/115). The daily events identified to carried out studies are standing or siting (*Shoaib et al., 2014*; *Li et al., 2016*; *Del Rosario et al., 2014*; *Incel & Ozgovde, 2018*; *Mafrur, Nugraha & Choi, 2015*; *Lee & Kwan, 2018*; *Capela et al., 2016*; *Hnoohom, Mekruksavanich & Jitpattanakul, 2017*; *Bort-Roig et al., 2019*), on foot moving (*Shoaib et al., 2014*; *Wang & Zhang, 2015*; *Arif et al., 2014*; *Spinsante et al., 2016*; *Yang et al., 2014*; *Aguiar et al., 2014*; *Del Rosario et al., 2014*; *Merchán-Baeza, González-Sánchez & Cuesta-Vargas, 2018*; *Incel & Ozgovde, 2018*; *Mafrur, Nugraha & Choi, 2015*; *Lee & Kwan, 2018*; *Solanas et al., 2015*; *Capela et al., 2016*), lying down (*Yang et al., 2014*; *Aguiar et al., 2014*; *Del Rosario et al., 2014*), going up and down the stairs (*Shoaib et al., 2014*; *Yang et al., 2014*; *Del Rosario et al., 2014*), and riding any motor vehicle (*Shoaib et al., 2014*; *Li et al., 2016*; *Trifan, Oliveira & Oliveira, 2019*; *Wan, Lin Kan & Wilson, 2017*; *Juen et al., 2014*). Additionally, other activities considered were skipping (*Trifan, Oliveira & Oliveira, 2019*), being motionless, surfing, cooking, dining and shopping activities (*Wan, Lin Kan & Wilson, 2017*). Those

activities are assessed by identifying and evaluating burnt calories, counting steps (*Chu et al., 2014*; *Tang, Guo & Chen, 2016*), diet (*Khedr & El-Sheimy, 2017*; *Luštrek et al., 2015*), to prevent from living sedentary identifying the motion (*Gu et al., 2017*), non exercise actions (*Madhushri et al., 2016*; *Lathia et al., 2017*; *Liu & Chan, 2016*), and number of hours spent and to be spent in walking and pulmonary diseases patient's movements (*Kelly, Curran & Caulfield, 2017*).

### Overall well-being

Some of the chosen publications looked into each of the health elements stated above separately, while others looked into multiple areas to acquire a better understanding of the consumers' overall well-being. Such systems detect sleep patterns, exercises activities, users' physical activities, and consciousness (*DeMasi et al., 2017*; *Lane et al., 2014*; *Aslam et al., 2016*; *Mo et al., 2015*; *DeMasi & Recht, 2017*) and position in order to fully comprehend and adjust their actions, or to improve development off lifestyle.

### Sleep

Only 7.9 percent (9/115) of the publications chosen used irregular night, sleep patterns, and sleep start and finish times to deduce users' sleep (*Cheffena, 2015*; *Huang et al., 2015*; *Saeb et al., 2017a*; *Montanini et al., 2018*; *Sarda et al., 2019*; *Lin et al., 2019*; *Nakano et al., 2014*). A few studies (*Staples et al., 2017*) looked at the relation among sleep models and schizophrenia.

### Student-specific monitoring systems

Monitoring student's lives was the focus of the research with a specific demographic behaviors (*Huang et al., 2016*; *Pulekar & Agu, 2016*; *Wang et al., 2014*; *Wang et al., 2015*; *Vhaduri, Munch & Poellabauer, 2016*; *Harari et al., 2017a*; *Chen et al., 2014*). Out of these, about 15.6 percent (18/115) of the studies selected created student-specific monitoring systems, in order to predict activities such as walking, running, jogging, sleeping, communication, and movement intensity (*Wang et al., 2014*; *Wang et al., 2015*; *Baras et al., 2016*), social phobia (*Saeb et al., 2017b*), psychological health (*Buck et al., 2019*; *Farhan et al., 2016*; *Yue et al., 2018*; *Boukhechba et al., 2018*), mobility, and attitudes (*Mafrur, Nugraha & Choi, 2015*; *Tseng et al., 2016*; *Hossain & Poellabauer, 2016*; *Vhaduri, Munch & Poellabauer, 2016*; *Harari et al., 2017a*). One study (*Chen et al., 2014*) presented a mechanism for anticipating students' shopping habits of food in their local region and offering healthier alternative options.

### Mental health

Mental health issues are another health-related topic that is well-studied in the publications chosen. Major mental health related diseases are studied due to usage of smart mobile phones such as anxiety (*Wang et al., 2016*; *Saeb et al., 2017b*), bipolar syndrome (*Matthews et al., 2016*; *Beiwinkel et al., 2016*; *Maxhuni et al., 2016*; *Wang et al., 2016*), schizophrenia (*Boukhechba et al., 2017*; *Ben-Zeev et al., 2016*), depression (*Ben-Zeev et al., 2016*; *Buck et al., 2019*; *Farhan et al., 2016*; *Yue et al., 2018*; *Boukhechba et al., 2018*), stressful situations (*Juen, Cheng & Schatz, 2015*; *Garcia-Ceja, Osmani & Mayora Ibarra, 2015*; *Ben-Zeev et al.,*

*2015*), psychotic deterioration (*Higgins et al., 2011*), and mood (*Pratap et al., 2019*; *Ben-Zeev et al., 2017*; *Servia-Rodríguez et al., 2017*; *Grünerbl et al., 2014*; *McNamara & Ngai, 2016*; *Zhang et al., 2018*; *Jeong & Breazeal, 2016*).

The research on consumers' typing habits and texting speed is an innovative way of assessing their emotions (*Higgins et al., 2011*). Evaluating the effect of users' exposure to surroundings (*Triguero-Mas et al., 2017*) used to determine the mental health by passive sensing. Two more studies (*Ellis et al., 2012*; *Wahle et al., 2016*) built their monitoring method, which included a recommendation system to help depressed individuals cope with their illness. Attendants also used the disability service system to get a review of circumstances faced by patients with depression (*Soares Teles et al., 2017*) or to inform psychiatrists and family and friends if patients with mood disorders exhibit unusual behavior (*Ng et al., 2014*).

### Cordiality

Cordiality is a level of interaction with the peer groups (friends, colleagues, family and others). It is recognized to have a significant impact on people's psychological distress levels. According to the majority of articles retrieved, Only a few factors have been considered to be allied, although it can improve the prediction in a better aspect. Healthy relationships among coworkers are shown to increase productivity (*Boonstra et al., 2017*), united families are more contented (*Ng et al., 2014*; *Sahiti et al., 2017*), and pupils cope better with their schoolwork when they are accompanied by friends (*Ng et al., 2014*). Exploring interaction patterns and nearby areas is one way of studying this health factor (*Luo et al., 2015*; *Sofia et al., 2016*). In two research works (*Singh, Goyal & Wu, 2018*; *Bati & Singh, 2018*), an intriguing strategy for exploiting the sensing capabilities of smartphones to predict users' risk-taking tendencies were explored (*Pulekar & Agu, 2016*).

## Self reports-traditional scales used to analyze mental health

Before releasing smartphone-based passive monitoring systems to the general public, it is necessary to do extensive testing to guarantee that users are engaged with the device. These scales are used to evaluate and compare results like behaviours, length of study and number of participants involved to access the study. The data were collected using a cross-sectional survey method, which entails gathering data at a single moment in time (*Lindner et al., 1999*). During one phase of data collection, the phenomena under inquiry are captured as they present themselves. In order to cross-validate and test the recent instrument's measurement, clinical assessment scales are used. Table 6 shows the major valuation scales used (*Polit & Hungler, 1993*; *Lindner et al., 1999*; *MacInnes, 2003*; *Bernard, 1998*). The data collected by smart mobile phones can be compared and validated using ground truth data. The sort of data utilised as base ground truth was disclosed in around 60.8 percent (70/115) of the evaluated papers, while the other research provided no relevant information. However, there are several drawbacks to this strategy, such as the fact that users may not always react precisely, resulting in inconsistent bipolar outcomes.

### Atlman self-rating for mania-ASRM

A set of five statements describes the primary signs of Maina as defined by the DSM-IV (*American Psychiatric Association, APA, 1994*). Individuals were directed to evaluate only one of the five statements from each group that best reflected their mindset or behaviour in the previous week, scored in severity from zero (not present) to four (present in severe degree). To enable for self-assessment of depressive episodes, three extra sets of five statements were included (*e.g.*, auditory and visual hallucinations, delusional thoughts). These questions were then reviewed and modified by other staff physicians who had experience with manic patients.

### Shortened general attitude and belief scale -SGABS

An elongated version of the General Attitude and Belief Scale (GABS), contains more than 50 items in the measuring scale (*Singh, Goyal & Wu, 2018*; *Malouff & Schutte, 1986*). It contains seven items of reasonableness along with one sub-scale (demand for trust, need for relief, need for support, need for attainment) (*Thomas & Bond, 2014*; *Cornet & Holden, 2018*).

### Irrational belief scale –IBS

To investigate the well-being, a self-report questionnaire on various factors of irrational behaviours are listed. About 20 items are to be answered and the total score was calculated by accumulating them together. Higher scores on the measure imply more illogical beliefs. The scale is frequently utilized in REBT research and clinical practice (*Thomas & Bond, 2014*; *Wertheim & Poulakis, 1992*; *As, 1983*).

### Hospital anxiety and depression scale –HADS

A fourteen-item fear and sadness assessment was separated into two subscales (*Goldberg, 1972*). The total score for each sub-scale ranges from 0 to 21, and respondents rate their answers on a four-point scale. Clinical significance is defined as a score of 11 or higher.

### General health questionnaire-28 –GHQ

Completion of self-analysis for measuring mental health disturbance by answering the questions (*Beck, 1979*). Its major focuses on items related to depression and anxiety are widely used in more clinical 439 research projects. In the survey, users are asked to rate the severity of their current experience based on a six-item severity scale. Scores of six or higher are indicative of overall health issues (*Bowling, 1991*).

### Beck depression inventory -BDI

The majority of depression predicting papers uses this BDI self-rating scale that consists of 21 questions. The total score range from 0 to 63, with which a higher score suggests depression and lower one exemplifies a good attitude. It is suggested that the inventory be used in clinical and research settings (*Beck, 1979*). Several methodological features of smartphone-based health monitoring systems should be considered. The most intriguing of these is the ability to discreetly and consistently collect well-being information about users without requiring them to change their everyday routines, transforming smartphones into a less intrusive and demanding tool than other health gadgets. Furthermore, mobile

phones are portable, less expensive, and efficient than other gadgets, and stay with users for a longer duration, making them utilize the smartphone as a familiar tool (*Anderson, Burford & Emmerton, 2016*). Since these crucial health related data from the inactive device should be preserved from others such as friends and the outside world.

## DISCUSSION

People may be hesitant to use smartphones for health monitoring because of the previously mentioned disadvantages. Users currently make rapid judgments about whether or not to use an app; as a result, their needs must be met completely. Consumers frequently judge apps based on their design initially. Generally, users require a user-friendly app and they were not willing to spend much time learning to use it. Battery usage and privacy are common concerns. In actuality, because these programs are often run in the background, consumers do not expect their battery level to decline significantly. Users may also delete apps if they have privacy concerns. Data acquired on a mobile is personal and should not be shared or accessed without permission. Commonly, people are not ready to share their health-related data on social media sites. But, at the same time, they may share it openly with physicians if they went for diagnosing health-related issues. Furthermore, consumers are comfortable utilising password-protected apps but are wary about putting in too much effort to create accounts. Moreover, an app's deactivation may be due to unreliable or inadequate outcomes or advice. Memory space used for application storage and working in the background is also an issue during normal smartphone operations (*Torous, Friedman & Keshavan, 2014*; *Carbonell, Michalski & Mitchell, 1983*).

Yet another critical factor to be considered is based on their present situation, people may receive a few numbers of calls, texts, emails and notifications with the majority of them being favourable recommendations. They find the ability to customize the frequency and timing of notifications to be appealing (*Torous, Friedman & Keshavan, 2014*). Users, on the other hand, are equally interested in defining and accomplishing personal goals. (*Dogan et al., 2017*; *Nguyen & Silva, 2016*).

In light of the aforementioned obstacles and potential issues, the created systems discussed in this comprehensive study still have some constraints that must be addressed in order to meet users' expectations and demands. It is critical for the validation of such monitoring systems to include a population sample that is representative of the target population for an extended period of time, in order to collect sufficient data and produce reliable results. From the above findings, about 83.4 percent (96/115) of the chosen papers tested their system with up to 250 participants, and their entire working observed for study purposes ranged up from one to four weeks in 16.5 percent (19/115) of the chosen papers, thus it looks like very short duration to guarantee realistic performance and ensure the customers' confidence in operating with the systems.

Additionally, the discussed models were suited for a limited population, which could lead to erroneous results when the system is used with a different population. Although 44.3 percent (51/115) of the selected papers proved that sensing can be passively done without disturbing the regular activities, it had to be retained in exclusive locations for effective

measurement such as pockets, purses, handbags and hands. According to the above study, smartphones can collect health-related data and offer users relevant feedback on health problems. Despite the increased interest and evolution, monitoring systems will require improvement in order to attract a wider range of users and achieve their expectations. In addition to the aforementioned demands and problems, the use of Mobile phones in health monitoring may generate further questions. Smartphones are currently utilised all around the world. The health monitoring systems may be beneficial to the elderly, smartphones may not be a simple or flexible tool for them but the younger generation is more familiar with them.

Furthermore, these systems may appeal to those with diagnosed ailments who have specific goals in mind, like observing lifestyles, regulating heartbeat rate, working of hormones, and reducing obesity, rather than people who have no clear goals in mind. Finally, once users are either comfortable with such systems or reached their individual goals, they may stop using the monitoring devices. The research presented shows how individuals can improve their health and well-being by monitoring many health dimensions exclusively using data collected from their smartphones. Most of the papers address unique disorders related to mental health, including depression, fear, anxiety, bipolar disorder, mood cornered problems, schizophrenia (*Faurholt-Jepsen, Bauer & Kessing, 2018*; *Rajagopalan et al., 2017*; *Dogan et al., 2017*; *Areán, Ly & Andersson, 2016*; *Þórarinsdóttir, Kessing & Faurholt-Jepsen, 2017*; *Nguyen & Silva, 2016*), heart related diseases (*Ko et al., 2015*), tension (*Nguyen & Silva, 2016*), sleep (*Ko et al., 2015*), and other physiologically health issues including obesity due to lack of physical exercises (*Thomas & Bond, 2014*), long-lasting diseases in older adults (*Cornet & Holden, 2018*), overall health (predicting disturbance in sleep (*Harari et al., 2016*), and mobility patterns (*Leigh & Flatt, 2015*). Figure 3 represents the selected publications in research fields. From the above figure, it is evident that due to the increase in the number of cases related to various issues in mental health and in general people are much more consciences about physical health. The smartphone was the most widely utilized technology and device in the evaluations (*Faurholt-Jepsen, Bauer & Kessing, 2018*; *Rajagopalan et al., 2017*; *Batra et al., 2017*; *Dennison et al., 2013*), although only some works perform it by gathering the sensors data (*Faurholt-Jepsen, Bauer & Kessing, 2018*; *Batra et al., 2017*; *Nguyen & Silva, 2016*; *Ko et al., 2015*; *Thomas & Bond, 2014*). Rarely in some circumstances, phones are utilized to persuade users to conduct ecological temporary assessments (*Batra et al., 2017*; *Areán, Ly & Andersson, 2016*; *Nguyen & Silva, 2016*), to deliver notifications apps of the smartphone are used (*Rajagopalan et al., 2017*; *Batra et al., 2017*; *Dogan et al., 2017*; *Areán, Ly & Andersson, 2016*; *Þórarinsdóttir, Kessing & Faurholt-Jepsen, 2017*; *Ko et al., 2015*), or to send brief messaging service suggestions to users (*Rajagopalan et al., 2017*; *Thomas & Bond, 2014*). Wearable gadgets, (*Batra et al., 2017*; *Þórarinsdóttir, Kessing & Faurholt-Jepsen, 2017*; *Ko et al., 2015*; *Cornet & Holden, 2018*) are also considered in peer-reviewed articles, as replacement of smart phones and due to the reputation of the embedding sensors in any daily usage components to monitor passively are tablets, accessories, health trackers and smartphone-linked devices (*Batra et al., 2017*; *Ko et al., 2015*; *Thomas & Bond, 2014*; *Kim & Lee, 2017*; *Cornet & Holden, 2018*). Unlike the other publications in this review, this one does not focus on a particular situation

or sensor. The aim of this study is to find various and possible physical condition-related variables which can be tracked using a phone, further determining how majority of sensing systems replace or supplement conventional diagnostic procedures.

# CURRENT CHALLENGES IN PASSIVE SENSING FOR MENTAL HEALTH RESEARCH

Frequent users anticipate monitoring systems to give them relevant data and recommendations regarding their actions (*Torous, Friedman & Keshavan, 2014*). Users are provided with well-being feedback (*Harari et al., 2017b*; *Anderson, Burford & Emmerton, 2016*) related to mental stress, well being and overall physical activities in order to alter their standard of living.

The use of smartphones in health monitoring appears to be a promising research subject. The current solutions have significant drawbacks that must be overcome in order for users to feel comfortable and confident utilising such systems. In reality, monitoring systems may be deemed unsettling, unpleasant, and intrusive, such as the use of smartphones for diagnostic evaluations, which have social, economic, and cultural limitations. Sick or socially vulnerable people have different behaviours in different scenarios. The current challenges in passive monitoring are broadly classified as shown in Fig. 5 such as issues in the monitoring system, end-users issues, and behavioural marker issues.

### Challenges in monitoring systems

The created systems discussed in this scoping review still have several flaws that must be addressed in order to meet the expectations and demands of users.

- validation of monitoring systems

    – These systems should be tested with a population sample that is highly representative of the target population for a sufficient period of time in order to collect enough data and produce as accurate results as possible. From the survey, 71.1 percent (84/115) of the selected papers tested their system with up to 50 participants, and 17.7 percent (21/115) of the selected papers tested their system for one to three weeks, which appears to be a short period to ensure reasonable results and ensure users' confidence in using available solutions.

    – Majority of the system designed so far are assessed on younger generations additionally outputs may not be much reliable (*Cheffena, 2015*; *Del Rosario et al., 2014*). Single population alone was targeted in the vast number of research works that may be overly customized, resulting in erroneous findings when the systems are utilised by other groups (*Chen et al., 2014*; *Beck, 1979*). Personalized models outperformed general models, according to other studies (*Huang et al., 2015*; *Juen, Cheng & Schatz, 2015*; *Saeb et al., 2017a*; *DeMasi et al., 2017*).

- Furthermore, 43.2 percent (51/115) of the papers chosen demand users to keep their smartphones close to them or to use them in a specified bodily position, such as their hand, chest, or trouser pocket. Other research required users to have their handsets near them (*Bowling, 1991*; *Huang et al., 2015*; *Ellis et al., 2012*) or to keep them turned on

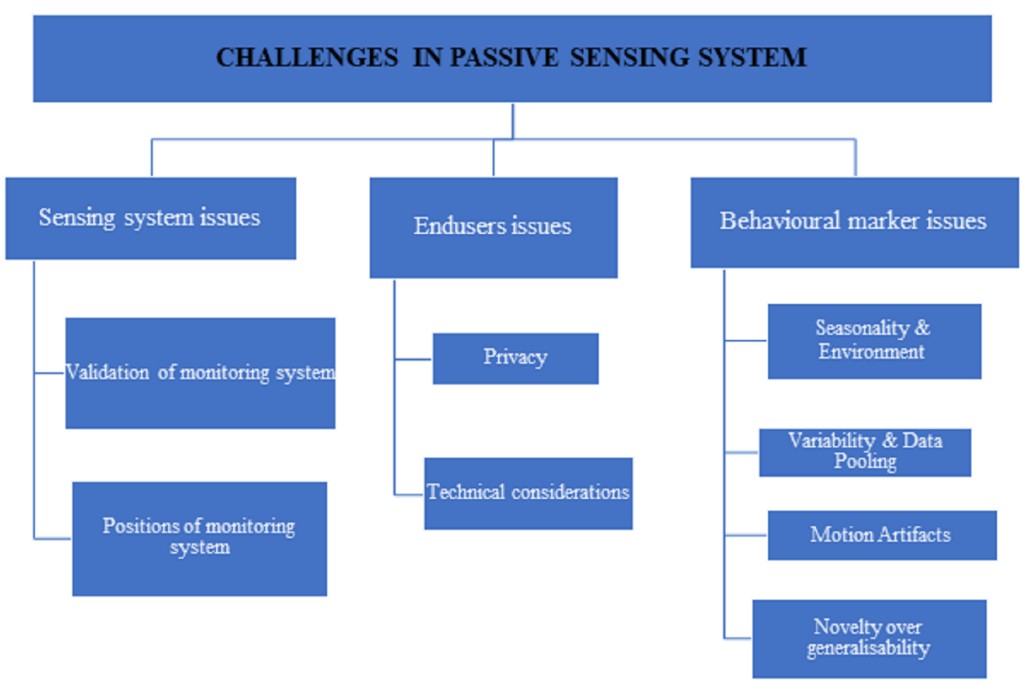

**Figure 5** The challenges in passive monitoring systems.

all the time to ensure that the system worked properly (*Boonstra et al., 2017*; *Staples et al., 2017*). Because smartphones become an invasive technology for users under certain situations, they may be rendered obsolete.

### End users issues

- In terms of privacy, 26.08 percent (30/115) of respondents indicated that privacy concerns could cause consumers to abandon their commitment. All those data are related to their personal these have to be kept secure were concentrated in (*Kelly, Curran & Caulfield, 2017*; *Wang et al., 2014*). In some studies users are not willing to save their personal data on cloud environment (*Ben-Zeev et al., 2017*; *Aslam et al., 2016*) or in their internal mobile devices by hashing techniques (*Huang et al., 2015*; *Boonstra et al., 2017*; *Aslam et al., 2016*; *Wang et al., 2014*), and there are sensor which contains certain concerns about privacy (*Juen, Cheng & Schatz, 2015*).
- Still on the subject of technological considerations, consumers anticipate that the software will not take up too much space or memory and that it will be able to run in the background without interfering with other smartphone functions.

### Behavioral marker issues

Seasonality and the environment are two other essential factors to consider. In the winter, for example, GPS and accelerometer data in Minneapolis will differ from that in Miami.

Variability in sensing data is due to the decentralized sources, including data types, people's characteristics, and diverse situations. Sensors in cellphones differ from one manufacturer to the next, from one model to the next, and from one version to the next,

impacting the data collected. The link between constructs or how people use measurement equipment may be influenced by their properties. For example, age may be linked to the number of social contacts, with older people having fewer contacts and desiring less contact, but it could also be linked to how social engagement is monitored using a phone (*e.g.*, older people are more likely to call and less likely to send text messages than younger people).

Finding a scientifically valid balance between identifying uniform variables, to making data pooling simple (*e.g.*, using the same questions) statistical methods is used to manage heterogeneity by providing similarity. It is challenging to pool data points if they are not identical. Since the field of personal sensing in mental health is not mature, some agreement may be possible on a core set of clinical assessment methods (EMA or self-report), which will allow for uniform anchoring of a wide range of sensor data as it evolves and changes over time and between research projects. A fully integrated assessment is complicated by the wide range of devices, sensors, and data permissions available.

Motion artefacts (MA) are common in EDA data gathered with wrist wearables. Variations in the pressure put on the EDA electrodes as a result of the wearable tightness, hand movements, or wrist rotation may cause severe data distortion. Many researchers have used approaches such as exponential smoothing, filtering, and adaptive denoising based on the wavelet transform to suppress artefacts in the past. MA suppression approaches, on the other hand, have the issue of indiscriminately filtering the entire time series data, resulting in distortions even in artifact-free areas. As a result, a new approach called MA detection was developed, which tries to efficiently encode expert knowledge on artefact recognition into a machine learning classifier model.

The technical novelty over generalisability research works appear to address the same behavioural marker often use different sensors, different sets of features, different methods of measuring behavioural markers, and different research designs (*e.g.*, giving people phones *versus* having them use their own, studying them for varying periods of time, or having varying numbers of participants excluded). The machine learning methods utilised differ, and the results or weightings, especially for group models, are not always comparable between research. As a result, studies looking at the use of machine-learning approaches to estimate behavioural markers show that it is possible under certain conditions; nevertheless, the dependability required for clinical use has yet to be established.

### Future work

Existing solutions on the other hand, have various flaws that must be solved in order to meet user expectations, such as privacy and battery issues. First, just a conceptual analysis was provided rather than a numerical analysis of the extent and type of the research. Second, a fully integrated assessment is complicated by the wide range of devices, sensors, and data permissions available. Thus, the sensor data from the monitoring system will be more accurate if a two-stage technique is used with an initial artefact detection phase followed by localised categorization depending on the target population. Finally, due to the limited sample size, different methodology, and varying research duration, we were

unable to use a systematic quality grading system or draw conclusions using quantitative meta-analysis.

## CONCLUSIONS

The number of people affected by mental illness has increased dramatically during the last decade richter. Importantly, many of these patients received inter care during COVID-19 as a result of an overburdened health service whose efforts were primarily committed to COVID-19. The global epidemic emphasises the need for contemporary digital tools in providing care when it is needed. In recent years, smartphone capabilities have enabled users to detect and track mental health issues, since it is available to individuals throughout the day. Technological advancements have made smartphones more accessible to users than traditional monitoring methods. A continuous stream of data is collected by the embedded sensors, resulting in minimal disruption to daily routines due to the collection of health data. This methodical review shows that sensing using a mobile phone and similar devices may create an authentic dataset, as seen by increased interest and awareness. Although there are a few significant areas where smartphone passive sensing contribute to users' well-being, there are many more that have yet to be discovered. While the accelerometer and GPS are predominately used, sensors alone are often used individually or combined. Only a few studies predict well-being based on the usage patterns and interactions made using smartphones. The smartphone evolved as an effective surveillance weapon because of its specific nature such as disconnected, and ubiquitous it allows continuous data accumulation from the users. Those data gathered by the smartphone can be made accessible by the medical experts, or caretakers in order to assist in the diagnosis and treatment of a variety of mental illnesses and even one step ahead may notify family members if they are far about apart.

### Funding
The authors received no funding for this work.

### Competing Interests
The authors declare there are no competing interests.

### Author Contributions
- Abinaya Gopalakrishnan conceived and designed the experiments, performed the experiments, analyzed the data, prepared figures and/or tables, authored or reviewed drafts of the article, and approved the final draft.
- Revathi Venkataraman conceived and designed the experiments, analyzed the data, authored or reviewed drafts of the article, and approved the final draft.
- Raj Gururajan conceived and designed the experiments, analyzed the data, authored or reviewed drafts of the article, and approved the final draft.
- Xujuan Zhou conceived and designed the experiments, analyzed the data, authored or reviewed drafts of the article, and approved the final draft.

- Rohan Genrich conceived and designed the experiments, analyzed the data, authored or reviewed drafts of the article, and approved the final draft.

## Data Availability

This is a literature review; there is no raw data.

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
