# Peer review of "Mobile phone enabled mental health monitoring to enhance diagnosis for severity assessment of behaviours: a review"

_PeerJ Computer Science, doi:10.7717/peerj-cs.1042_

## Round 0.1 · original submission · Major Revisions

The paper has touched a very interesting area, and the contribution is obvious, however some major commenters related to the validity of the results and figures must be fixed before publication in PeerJ Computer Science.

·

Basic reporting

The article meets the appropriate standards. Extensive references, the writing is clear and tables and figures are adequate.

Experimental design

The study design is correct. But it is necessary to explain in more detail the selection criteria of the chosen literature. It should also go deeper into the characteristics of the non-selected literature.
They must justify the selective criteria regarding the databases used and those not used. For example, why haven't you used The Web of Science or Scopus?
And offer information on the name of the publications with articles on the subject, with data on the number of publications per year, etc.

Validity of the findings

It is necessary to compare results with other similar existing studies. Emphasizing the coincidences and divergences on the results obtained. A critical reflection on the limitations of smartphones and passive devices to analyze personal health data is also necessary. Attention to bioethical limits, privacy, and confidentiality. Do you start from the fact that the smartphone user freely allows being monitored?
The use of smartphones for diagnostic evaluation has social, economic, and cultural limitations. Authors should keep this fact in mind. It is not possible to generalize passive monitoring to produce data in athletes when we refer to data monitoring in cases of people with medical pathologies or social inclusion problems. Sick or socially vulnerable people have different behaviors.
I suggest a greater emphasis on the discussion of the results and the limitations of the study carried out.

Reviewer 2 ·

Basic reporting

1. Literature in the field of public health has moved away from using the term “mental disorder” and towards “mental health issues” to avoid stigmatizing those affected. I would recommend adapting this language (e.g., in the abstract).
2. Kindly review the full manuscript for grammatical errors and sentence structure. In some areas the language is inconsistent (i.e., use of terms depression vs. psychological health issues) and can be simplified (i.e., lines 96-97).
3. The results organization is unclear and difficult to follow. The manuscript would benefit from clearly outlining the scoping review goal and specific sub-questions to enable this format to be followed in the results.
4. Overall, the manuscript text can be tightened to align more closely with the subheadings and avoid repetition throughout. For example, the purpose of digital tools for mental health assessment does not need to be repeated in the ‘Search Strategy’ subsection as this has been established in the Introduction.
5. Typically scoping reviews include a review of the grey literature. Given the focus on peer-reviewed literature, this could be categorized as a systematic review.

Experimental design

1. References are required in the first paragraph (lines 89-91).
2. The search strategy section can be more clearly organized to remind the reader of the key research question for the scoping review, search parameters (date range, databases), and inclusion and exclusion criteria. Currently information like the timeline (Jan 2015-Dec 2020) is repeated twice.
3. The additional rationale provided in the first few sentences is already in the introduction, and is not required again in this section.
4. Please note how many reviewers participated in the article screening process, and how/if any disagreements in the shortlist were resolved (i.e. consensus exercise).

Validity of the findings

1. Line 130 is unclear, what is “explicit human contact with the monitoring system?”
2. Please describe Figure 2 in greater detail. How was the framework compiled? Why is it organized as such? What do the acronyms used under Self-Report stand for?
3. The results can be organized for a clearer presentation. Additional text (i.e., lines 142-145, 288-290) distract from the results and should only be noted in the discussion section.
4. Please bold the first row in all Tables to distinguish the column headers.

Additional comments

Abstract:
1. The second sentence is unclear, how does the severity of mental health issues lead to observations of people through smartphones? The point about potential for smartphone-based data collection in the abstract can be communicated more concisely to improve clarity.

Introduction
1. The introduction requires references for many of the statements made in the first paragraph. Please review lines 36-53 and include references throughout.
2. Line 84 notes “estimating depression” however the introduction suggests that a range of psychiatric conditions are included in this review. Please refine the language based on the scoping review focus.
3. Two goals are mentioned in the Introduction (lines 68 and again on lines 80-81). This section would benefit from organizing to indicate perhaps the overarching goal (i.e., to develop a framework) informed by a scoping review. The text currently framed as the scoping review goal could be reformatting as the primary research question of the review. The list provided on lines 72-78 can be reformatted as objectives.

Discussion
1. Tables and figures are typically not placed in the Discussion section. Please move Tables 9 and 10 to the Results section and revise the discussion accordingly.
Conclusions
1. The conclusion section makes no mention of mental health issues which is meant to be the focus of this systematic review.

---

## Round 0.2 · Major Revisions

The paper needs more improvements. Please highlight the contribution and validity of your contribution. You must make it clear to the reader the new information and details you are showing.

Reviewer 3 ·

Basic reporting

What doe it mean by the statement at lines 93-94 and a number of research articles selected from the various databases to “fine” the research work. Please carefully read the article and remove these major mistakes.
Table 1 seems to be ambiguous, can you elaborate on how this table has been formed?
Your search query resulted in 943 manuscripts, but I am unable to find any automatic way to filter out these articles, have you read them manually and then selected the shortlisted articles or what? Please explain
Please write the complete name of figures in texti e.g. Fig might not be appropriate. At some places, it is complete e.g at line 138. Please make sure it should be consistent.
Please explain the steps in figure 1, the data flow diagram. All the steps were performed manually?
The literature review article should be a comprehensive study of relevant studies and their findings and recommendations for the readers. Unfortunately, these things seem to be lacking in the current article
There are lots of typos and grammatical mistakes; some of them are as follows

This physiological issues needs to be consistently observed on the people, could be.
These physiological issues need to be consistently observed in the people ….


Thus, the aim of this comprehensive work concentrates on the literature work done so far in the prediction of mental heath issues, what is “heath” in this one ?

Experimental design

What doe it mean by the statement at lines 93-94 and a number of research articles selected from the various databases to “fine” the research work. Please carefully read the article and remove these major mistakes.
Table 1 seems to be ambiguous, can you elaborate on how this table has been formed?
Your search query resulted in 943 manuscripts, but I am unable to find any automatic way to filter out these articles, have you read them manually and then selected the shortlisted articles or what? Please explain
Please write the complete name of figures in texti e.g. Fig might not be appropriate. At some places, it is complete e.g at line 138. Please make sure it should be consistent.
Please explain the steps in figure 1, the data flow diagram. All the steps were performed manually?
The literature review article should be a comprehensive study of relevant studies and their findings and recommendations for the readers. Unfortunately, these things seem to be lacking in the current article
There are lots of typos and grammatical mistakes; some of them are as follows

This physiological issues needs to be consistently observed on the people, could be.
These physiological issues need to be consistently observed in the people ….


Thus, the aim of this comprehensive work concentrates on the literature work done so far in the prediction of mental heath issues, what is “heath” in this one ?

Validity of the findings

What doe it mean by the statement at lines 93-94 and a number of research articles selected from the various databases to “fine” the research work. Please carefully read the article and remove these major mistakes.
Table 1 seems to be ambiguous, can you elaborate on how this table has been formed?
Your search query resulted in 943 manuscripts, but I am unable to find any automatic way to filter out these articles, have you read them manually and then selected the shortlisted articles or what? Please explain
Please write the complete name of figures in text e.g. Fig might not be appropriate. At some places, it is complete e.g at line 138. Please make sure it should be consistent.
Please explain the steps in figure 1, the data flow diagram. All the steps were performed manually?
The literature review article should be a comprehensive study of relevant studies and their findings and recommendations for the readers. Unfortunately, these things seem to be lacking in the current article
There are lots of typos and grammatical mistakes; some of them are as follows

This physiological issues needs to be consistently observed on the people, could be.
These physiological issues need to be consistently observed in the people ….


Thus, the aim of this comprehensive work concentrates on the literature work done so far in the prediction of mental heath issues, what is “heath” in this one ?

Additional comments

this paper needs serious level improvements for re-consideration

---

## Round 0.3 · Minor Revisions

The author has responded to most of the comments provided by the reviewers, However, still some comments must be addressed.

Reviewer 3 ·

Basic reporting

the author have made no change in Table1 , as track version and updated version is not showing at least. my question was to make it clear, they wrote in rebuttal but makes no changes in the actual table

???Thanks for your comments. We used search queries to retrieve 6 databases. Figure 1 depicts the steps we use followed to include the articles which meet the selection criteria. Please see Figure 1 for more details., another ambiguous answer, the actual comment was
Your search query resulted in 943 manuscripts, but I am unable to find any automatic way to filter out these articles, have you read them manually and then selected the shortlisted articles or what? Please explain

manual scanning of 963 articles seems to be a very hectic and tedious task, and almost impossible
it is difficult to judge why the authors have added RELEVANT WORKS section ? figure 4 is not properly cited
please re-structure the overall document in a proper article hierarchy and then resubmit

Experimental design

the author have made no change in Table1 , as track version and updated version is not showing at least. my question was to make it clear, they wrote in rebuttal but makes no changes in the actual table

???Thanks for your comments. We used search queries to retrieve 6 databases. Figure 1 depicts the steps we use followed to include the articles which meet the selection criteria. Please see Figure 1 for more details., another ambiguous answer, the actual comment was
Your search query resulted in 943 manuscripts, but I am unable to find any automatic way to filter out these articles, have you read them manually and then selected the shortlisted articles or what? Please explain

manual scanning of 963 articles seems to be a very hectic and tedious task, and almost impossible
it is difficult to judge why the authors have added RELEVANT WORKS section ? figure 4 is not properly cited
please re-structure the overall document in a proper article hierarchy and then resubmit

Validity of the findings

the author have made no change in Table1 , as track version and updated version is not showing at least. my question was to make it clear, they wrote in rebuttal but makes no changes in the actual table

???Thanks for your comments. We used search queries to retrieve 6 databases. Figure 1 depicts the steps we use followed to include the articles which meet the selection criteria. Please see Figure 1 for more details., another ambiguous answer, the actual comment was
Your search query resulted in 943 manuscripts, but I am unable to find any automatic way to filter out these articles, have you read them manually and then selected the shortlisted articles or what? Please explain

manual scanning of 963 articles seems to be a very hectic and tedious task, and almost impossible
it is difficult to judge why the authors have added RELEVANT WORKS section ? figure 4 is not properly cited
please re-structure the overall document in a proper article hierarchy and then resubmit

Additional comments

the author have made no change in Table1 , as track version and updated version is not showing at least. my question was to make it clear, they wrote in rebuttal but makes no changes in the actual table

???Thanks for your comments. We used search queries to retrieve 6 databases. Figure 1 depicts the steps we use followed to include the articles which meet the selection criteria. Please see Figure 1 for more details., another ambiguous answer, the actual comment was
Your search query resulted in 943 manuscripts, but I am unable to find any automatic way to filter out these articles, have you read them manually and then selected the shortlisted articles or what? Please explain

manual scanning of 963 articles seems to be a very hectic and tedious task, and almost impossible
it is difficult to judge why the authors have added RELEVANT WORKS section ? figure 4 is not properly cited
please re-structure the overall document in a proper article hierarchy and then resubmit

---

## Round 0.4 · accepted · Accept

The author has responded to the reviewers comments. All comments have been tackled by the author.